# Development of a New Nomogram Including Neutrophil-to-Lymphocyte Ratio to Predict Survival in Patients with Hepatocellular Carcinoma Undergoing Transarterial Chemoembolization

**DOI:** 10.3390/cancers11040509

**Published:** 2019-04-10

**Authors:** Young Eun Chon, Hana Park, Hye Kyung Hyun, Yeonjung Ha, Mi Na Kim, Beom Kyung Kim, Joo Ho Lee, Seung Up Kim, Do Young Kim, Sang Hoon Ahn, Seong Gyu Hwang, Kwang-Hyub Han, Kyu Sung Rim, Jun Yong Park

**Affiliations:** 1Department of Internal Medicine, Institute of Gastroenterology, CHA Bundang Medical Center, CHA University, Seongnam 13496, Korea; nachivysoo@chamc.co.kr (Y.E.C.); hnpark80@gmail.com (H.P.); yeonjung.ha@chamc.co.kr (Y.H.); mina2015@cha.ac.kr (M.N.K.); ljh0505@cha.ac.kr (J.H.L.); sghwang@cha.ac.kr (S.G.H.); ksrimmd@cha.ac.kr (K.S.R.); 2CHA Bundang Liver Center, CHA Bundang Hospital, Seongnam 13496, Korea; 3Department of Internal Medicine, Institute of Gastroenterology, Yonsei University College of Medicine, Seoul 03722, Korea; hyunhk0703@yuhs.ac (H.K.H.); beomkkim@yuhs.ac (B.K.K.); ksukorea@yuhs.ac (S.U.K.); dyk1025@yuhs.ac (D.Y.K.); ahnsh@yuhs.ac (S.H.A.); gihankhys@yuhs.ac (K.-H.H.); 4Yonsei Liver Center, Severance Hospital, Seoul 03722, Korea

**Keywords:** hepatocellular carcinoma, transarterial chemoembolization, neutrophil-to-lymphocyte ratio, risk-prediction model

## Abstract

The neutrophil-to-lymphocyte ratio (NLR) has recently been reported to predict the prognosis of hepatocellular carcinoma (HCC). We explored whether NLR predicted the survival of patients with HCC undergoing transarterial chemoembolization (TACE), and developed a predictive model. In total, 1697 patients with HCC undergoing TACE as first-line therapy at two university hospitals were enrolled (derivation set n = 921, internal validation set n = 395, external validation set n = 381). The tumor size, tumor number, AFP level, vascular invasion, Child–Pugh score, objective response after TACE, and NLR, selected as predictors of overall survival (OS) via multivariate Cox’s regression model, were incorporated into a 14-point risk prediction model (SNAVCORN score). The time-dependent areas under the receiver-operating characteristic curves for OS at 1, 3, and 5 years predicted by the SNAVCORN score were 0.812, 0.734, and 0.700 in the derivation set. Patients were stratified into three risk groups by SNAVCORN score (low, 0–4; intermediate, 5–9; high, 10–14). Compared with the low-risk group, the intermediate-risk (HR 3.10, *p* < 0.001) and high-risk (HR 7.37, *p* < 0.001) groups exhibited significantly greater mortality. The prognostic performance of the SNAVCORN score including NLR in patients with HCC treated with TACE was remarkable, much better than those of the conventional scores. The SNAVCORN score will guide future HCC treatment decisions.

## 1. Introduction

Hepatocellular carcinoma (HCC) is a common fatal cancer that is the third leading cause of cancer deaths worldwide [1,2]. Despite progress in various treatment modalities, prognosis remains very poor because few patients are candidates for radical treatments, such as resection or liver transplantation [3,4]; many patients are initially diagnosed with intermediate to advanced HCC, and some with severe hepatic dysfunction cannot endure curative treatment [5,6]. In patients diagnosed with above early stage HCC, with Child–Pugh class A or B disease, transarterial chemoembolization (TACE) has been used widely as treatment since the 1980s [7,8,9]. Despite the proven survival benefits, prognosis after TACE varies because patients with intermediate-stage HCC are heterogeneous in terms of tumor burden, tumor markers, liver function, and performance status [10]. Therefore, factors predictive of overall survival (OS) in patients undergoing TACE have been investigated extensively to enable selection of patients who will benefit from TACE. Such factors include tumor factors (tumor number, size, location, and vascularity; portal vein invasion) [11,12]; patient factors such as age, Child–Pugh class, alpha-fetoprotein (AFP) level, and performance status have been also suggested as predictors [13,14]. Moreover, a complete tumor response after TACE, defined as the disappearance of any intratumoral arterial enhancement on localized lipiodol administration, is an important predictor of prognosis [15]. Based on such predictors, several models assessing prognosis after TACE have been defined [ART score (the Assessment for Retreatment with TACE), SNACOR model (tumor Size, tumor Number, AFP level, Child–Pugh class, Objective Response after TACE), ABCR score (AFP level, BCLC stage, Child–Pugh class, Response after TACE), HAP (Hepatoma Arterial-embolisation Prognostic) score, and mHAP II (modified HAP II) score] [16,17,18,19,20]. However, their efficacies have not been compared extensively in real-life settings. 

In cancer patients, many inflammatory cytokines and mediators induce systemic inflammation, which is regarded as a host immune response [21]. The neutrophil-to-lymphocyte ratio (NLR), a biomarker of inflammation, is prognostically valuable in patients with many solid tumors, including ovarian and colon cancers, and HCC [22,23,24]. In several previous studies, elevated NLRs were correlated with poor survival in patients with HCC undergoing TACE [25,26,27]. However, the small samples of those studies precluded confirmation of the hypothesis. Here, we explored the prognostic utility of the NLR in a large cohort of patients with HCC undergoing TACE, and generated a prognostic TACE model. We compared the prognostic utility of the new model with those of conventional models.

## 2. Results

### 2.1. Basline Characteristics

The baseline characteristics of the derivation set are shown in Table 1. The mean patient age was 68 years and 76% of patients were male. Hepatitis B virus infection was the most prevalent HCC etiology (70.4%); most (88.3%) patients evidenced reserve liver function (Child–Pugh class A). The median tumor size was 5.0 cm, and 425 (74.2%) patients had single tumors. The Barcelona clinic liver cancer (BCLC) stage was A in 425 (46.2%) patients, B in 352 (38.2%), and C in 144 (15.6%) patients. The mean NLR was 3.5. 

The baseline characteristics of the internal validation set were similar to those of the derivation set, except that the former patients were younger (66 vs. 68 years, *p* < 0.001), had a lower mean aspartate minotransferase (AST) (45 vs. 57 IU/L, *p* < 0.001), alanine transaminase (ALT) (34 vs. 41 IU/L, *p* = 0.003), NLR (2.8 vs. 3.5, *p* < 0.001). Compared with patients in the derivation set, those in the external validation set were younger (mean age 60 vs. 68 years, *p* < 0.001), had smaller HCCs (median tumor size 3.5 vs. 5.0 cm, *p* = 0.042), had multiple HCCs (32.8% vs. 25.8%, *p* < 0.001), and had early-stage HCC (BCLC A stage, 62.2% vs. 46.2%, *p* < 0.001). In addition, patients in the external validation set were of advanced Child–Pugh class (B, 20.5% vs. 11.7%, *p* < 0.001) and had a lower mean NLR (2.3 vs. 3.5, *p* < 0.001). The median survival time in the derivation set was 45.7 (95% CI 40.2–51.1) months, and those in the internal and external validation sets were 41.0 months (95% CI 2.6–79.4 months, *p* = 0.677) and 34.0 months (95% CI 28.2–40.0 months, *p* = 0.412) respectively, thus not significantly different (log-rank test).

### 2.2. TACE Responses

All patients underwent conventional TACE. Of the various TACE procedures, 782 (84.9%) used adriamycin and 139 (15.1%) used cisplatin. After the first TACE, 465 (50.4%) patients evidenced complete response (CR), 262 (28.5%) partial response (PR), 115 (12.5%) stable disease (SD), and 79 (8.6%) showed progressive disease (PD). In 262 patients who showed PR after first TACE, 121 (46.2%) patients underwent repeated TACE, 95 (36.2%) patients underwent concurrent chemoradiation therapy, 12 (4.6%) patients underwent radiation therapy, eight (3.1%) patients underwent sorafenib, eight (3.1%) patients underwent liver resection, eight (3.1%) patients underwent radio frequency ablation (RFA), eight (3.1%) patients underwent intra-arterial chemotherapy, and two (0.8%) patients had conservative care. Among all patients with PR, 76 (29%) patients achieved CR after this additional therapy. In 194 patients who showed SD or PD after first TACE, 66 (34.0%) patients underwent repeated TACE, 60 (30.9%) patients underwent concurrent chemoradiation therapy, 21 (10.8%) patients underwent sorafenib, 20 (10.3%) patients had conservative care, 12 (6.2%) patients underwent radiation therapy, 11 (5.7%) patients underwent intra-arterial chemotherapy, and four (2.1%) patients had RFA. Patients had undertaken mean of 1.4 sessions of TACE for the tumors.

### 2.3. Prognostic Factors for OS and Calculation of Risk Scores

On univariate analysis, the baseline serum AFP level (≥200 vs. <200 ng/mL), tumor size (>3 cm vs. ≤3 cm), tumor number (>3 vs. ≤3), vascular invasion, BCLC stage (C or B vs. A), Child–Pugh score (6 or 7–8 or 9 vs. 5), treatment response after the first TACE (SD+PD vs. CR+PR), serum albumin, total bilirubin, and baseline NLR (≥ 5 vs. <5) were predictors of poor OS in the derivation set (Appendix A). No on-treatment variable, including the AFP level, Child–Pugh score, NLR, or AST or ALT level 1 month after TACE, nor any change in these variables from baseline to 1 month after TACE, was significantly predictive of OS. 

After excluding BCLC stage, serum albumin, and total bilirubin to avoid the potential effect of multi-collinearity, multivariate analysis was performed using the seven variables (tumor Size, tumor Number, AFP level, Vascular invasion, Child–Pugh score, Objective Response after TACE, Neutrophil-to-lymphocyte ratio). Finally, the seven variables were incorporated into a risk prediction model; SNAVCORN scores ranged from 0 to 14, with weighting depending on the ratio of the regression β-coefficient (Table 2).

### 2.4. Predictive Performance of the SNAVCORN Score

In the derivation set, the time-dependent AUROCs of the SNAVCORN score for OS at 1, 3, and 5 years were 0.812 (95% CI 0.769–0.856), 0.734 (95% CI 0.396–0.770), and 0.700 (95% CI 0.663–0.737), respectively (Table 3). The 1-, 3-, and 5-year AUROCs of the SNAVCORN score for OS in the internal validation set were 0.868 (95% CI 0.795–0.942), 0.742 (95% CI 0.650–0.833), and 0.745 (95% CI 0.658–0.832), respectively, and those for OS in the external validation set were 0.801 (95% CI 0.716–0.885), 0.789 (95% CI 0.726–0.853), and 0.725 (95% CI 0.655–0.796), respectively (Table 3).

When we compared OS prediction at 1, 3, and 5 years (SNAVCORN score vs. ART, SNACOR, ABCR, HAP, and mHAP II scores), AUROCs for the SNAVCORN score were higher than those for the conventional risk prediction models (all *p* < 0.05; Table 4). The Kaplan–Meier OS curves for these conventional risk prediction models are shown in Appendix A.

### 2.5. Risk Prediction by Group

The SNAVCORN score stratified patients into three risk groups: low (score 0–4), intermediate (score 5–9), and high (score 10–14) risk. In the derivation set, the median OS was significantly higher in the low-risk group at 63.3 (95% CI 52.7–74.5) months, followed by the intermediate-risk (16.3 (95% CI 12.0–20.6) months) and high-risk (8.3 (95% CI 7.0–9.6) months) groups (log-rank test, *p* < 0.001; Figure 1a). The intermediate-risk (HR 3.10, 95% CI 2.5–3.8, *p* < 0.001) and high-risk (HR 7.37, 95% CI 4.10–13.25, *p* < 0.001) groups were at significantly greater risk of death compared with the low-risk group.

In the internal validation set, the low-risk group tended to have the longest median OS, at 65.0 (95% CI 21.6–108.4) months, consistent with data from the derivation set: the median OS in the intermediate-risk group was 19.6 (95% CI 7.6–31.5) months and that in the high-risk group was 5.6 (95% CI 0.5–10.7) months (log-rank test, *p* < 0.001; Figure 1b). The intermediate-risk (HR 5.78, 95% CI 2.81–11.84, *p* < 0.001) and high-risk (HR 22.37, 95% CI 8.17–61.23, *p* < 0.001) groups were at significantly greater risk of death compared with the low-risk group. 

Similarly, in the external validation set, the low-risk group exhibited the longest median OS of 47.7 (95% CI 32.5–62.9) months, followed by the intermediate-risk (16.2 (95% CI 9.9–22.4) months) and high-risk (7.1 (95% CI 3.5–10.7) months) groups (log-rank test, *p* < 0.001; Figure 1c). In the external validation set, the intermediate-risk (HR 2.89, 95% CI 1.99–4.18, *p* < 0.001) and high-risk (HR 6.47, 95% CI 3.38–12., *p* < 0.001) groups were at significantly greater risk of death compared with the low-risk group.

## 3. Discussion

TACE is one of the most commonly used palliative treatments for HCC. In clinical practice, patients undergoing TACE exhibit various HCC stages and extents of liver function, and the treatment outcomes in terms of tumor control rate and survival are very diverse. Therefore, patients who may enjoy survival benefits after TACE must be identified accurately; TACE should be performed selectively. Here, we developed a novel risk prediction model (the SNAVCORN score) predicting the prognosis of patients with HCC after TACE using simple, but significant, variables. Notably, the SNAVCORN model is unique in that it features the NLR, a marker of inflammation, and is more accurate than earlier predictive models.

Baseline tumor factors, including the BCLC stage and AFP level, which reflect biological features of HCC, and liver function assessed using the Child–Pugh score predicts OS after TACE. However, the Child–Pugh score and AST and ALT levels 1 month after TACE did not differ significantly from the baseline levels (data not shown); such changes were thus not predictive of OS. Of the many possible variables, a good radiological response after TACE was the only on-treatment variable predictive of OS. Thus, we created a simple but powerful HCC predictive model consists of **six** baseline factors and one on-treatment factor (the radiological response after TACE).

Infection and subsequent chronic inflammation are known to be critical in terms of tumor proliferation and invasion triggering the neoplastic process [28]. In addition, inflammation is associated with the production of various cytokines and signaling factors stimulating tumor migration and metastasis [29]. Chronic inflammation induced by hepatitis B or hepatitis C virus infection is pivotal in terms of HCC predisposition [30]. Therefore, the inhibition of viral replication or viral elimination using anti-viral agents is a fundamental precaution for prevention of HCC by reducing chronic inflammation.

Traditionally, the level of serum C-reactive protein (CRP), an acute-phase marker of systemic inflammation synthesized by hepatocytes, has been a poor prognostic marker in patients with various cancers, including HCC [31,32,33,34]. Jun et al. concluded that the serum CRP level was associated significantly with increased 10-month mortality in patients with large (>5 cm) HCCs undergoing TACE [35]. The NLR is another marker of the systemic inflammatory response, correlating with tumor progression, metastasis, and the clinical outcomes of various cancers [22,23]. Specifically, the neutrophil count (reflecting the inflammatory microenvironment) may be closely associated with cancer proliferation, survival, angiogenesis, metastasis, and subversion of the adaptive immune response [36]. On the other hand, lymphocyte numbers reflect the extent of host immunity and are associated with cancer suppression, survival, and reduced progression [37]. Thus, the NLR may represent the critical balance between chronic inflammation contributing to tumor growth and the extent of host anti-tumor immunity. Elevated NLRs are correlated with poor OS in patients with HCC undergoing liver transplantation, curative resection, radiofrequency ablation, and TACE [24,25,38,39].

In the present era of cancer immunotherapy, the NLR serves as a prognostic marker during various cancer treatments. For example, renal cell carcinoma is immunogenic in nature and the therapeutic use of immune checkpoint inhibitors was approved several years ago [40]. Lalani et al. showed that the NLR 6 weeks after treatment independently predicted OS and progression-free survival in patients with metastatic renal cell carcinoma subjected to a programmed death-1 or programmed death-ligand 1 based immune checkpoint blockade [41]. In patients with advanced HCC, sorafenib, an oral multi-kinase inhibitor, is the treatment of choice, and significantly prolongs OS [42]. Bruix et al. found that a lower NLR was significantly prognostic of better survival in a sorafenib-treated group, and a significant predictor of better treatment outcomes [43]. Recently, several immune checkpoint and multi-kinase inhibitors have been introduced as first-line therapeutics (lenvatinib) [44], or as second-line treatments after sorafenib (regorafenib, nivolumab, cabozantinib, and ramucirumab) for advanced HCC [45,46,47,48,49]. Investigation of the prognostic utility of the NLR (an immune system marker) when these new agents are given to patients with advanced HCC will be useful.

When designing a prognostic model, SNAVCN score consisting of six baseline predictors was simpler, however, addition of the TACE response (SNAVCORN) significantly improved predictive performance. Moreover, SNAVCORN score with addition of the easily calculable NLR, showed significantly more powerful predictive performance than SNAVCOR score (Appendix A). This new SNAVCORN score better predicted OS than did the conventional predictive models. The AUROCs at 1, 3, and 5 years were higher than those of the ART, SNACOR, ABCR, HAP, and mHAP II models. In addition, we demonstrated the predictive performance of the SNAVCORN score in an external cohort of patients with HCCs of different stages and with various levels of liver function. Thus, the SNAVCORN score may be applied safely to evaluate many other cohorts of patients with HCC undergoing TACE. Another major advantage of our study is the large sample. Previous studies identified a role for the NLR in fewer than 200 patients, but we enrolled more than 1000 patients when creating our NLR-based predictive model. Finally, a unique advantage of our study is that we are the first to construct a survival prediction model that includes the NLR, and immunological and inflammatory prognostic factors, for patients with HCC undergoing TACE.

Our study has certain limitations. First, the work was retrospective in nature; the data may be somewhat heterogeneous. Second, because various secondary treatments (TACE, chemoradiation, radiation, sorafenib, liver resection, RFA, intra-arterial chemotherapy, and conservative care) were performed after the first TACE, the effect of these subsequent treatments for the OS was hard to be analyzed. However, our aim was to predict the OS of all patients before and immediately after the first TACE, which was the initial treatment. For SNAVCORN high-risk patients, physicians must be careful when selecting and performing additional treatment, as the future OS will be short no matter what secondary therapy is chosen. Third, the mechanism connecting the increased NLR to poor prognosis remains unclear. More studies are needed to elucidate the immunological and molecular mechanisms.

## 4. Materials and Methods 

### 4.1. Patients

Patients newly diagnosed with HCC and undergoing TACE as first-line therapy between 2008 and 2017 at Severance Hospital, Yonsei University College of Medicine, were enrolled retrospectively. Of 1531 patients who received conventional TACE, 215 were excluded because of: (1) major vascular invasion (n = 97), (2) extrahepatic metastases (n = 5), (3) Child–Pugh class C (n = 10), (4) liver transplantation (n = 35), and (5) combination of TACE with another treatment modality (n = 68). Data from a total of 1316 patients were finally analyzed, and were divided randomly into a derivation set (70%, n = 921) and an internal validation set (30%, n = 395). We also collected data on 483 patients newly diagnosed with HCC who received TACE as first-line therapy between 2008 and 2017 at CHA Bundang Hospital, CHA University. The median follow-up durations were 34.5 (13.0–61.4) months in the derivation set, 12.5 (7.3–49.0) months in the internal validation set, and 29.9 (13.0–57.9) months in the external validation set. After application of the exclusion criteria described above, 381 patients were enrolled to constitute the external validation cohort. The study was approved by the institutional review boards of Severance Hospital and CHA Bundang Hospital (protocol codes: 4-2017-1116 and 2016-03-039-016). The study was conducted in accordance with all relevant ethical guidelines of the 1975 Declaration of Helsinki.

### 4.2. Endpoints and Definitions

The primary endpoint was OS, defined as the interval between the date of TACE and the date of death or last follow-up. HCC was diagnosed histologically or radiologically in accordance with the guidelines of the American Association for the Study of Liver Diseases or the European Association for the Study of the Liver [12,50]. We recorded baseline age; sex; liver disease etiology; the levels of AST, ALT, total bilirubin, albumin, and AFP; the prothrombin time (PT); the Child–Pugh score; tumor size and number; the BCLC stage; and the NLR. One month after TACE, we checked AST, ALT, total bilirubin, albumin, and AFP levels; the PT; the Child–Pugh score; the NLR; and tumor response via follow-up CT. Tumor response was assessed using the modified Response Evaluation Criteria in Solid Tumors and classified as CR, PR, SD, and PD [51].

### 4.3. TACE and Follow-Up

Selective angiography of the superior mesenteric, celiac, and common hepatic arteries was performed to evaluate vessel anatomy and tumor vascularity. A mixture of 30–50 mg adriamycin or cisplatin (2 mg/kg body weight) with 5 mL iodized oil contrast medium (lipiodol) was infused selectively through a 5-Fr catheter into subsegmental or segmental branches of the feeding arteries. Embolization was then performed with the aid of gelatin sponge pledgets. Follow-up computed tomography (CT) was performed 1 month later to evaluate the tumor response. In patients who showed CR at CT scan 1 month after first TACE, were regularly followed with CT scan taken every 3–6 months. In patients showing PR, SD, or PD at CT scan 1 month after first TACE, subsequent treatments (i.e., repeated TACE, radio frequency ablation, intra-arterial chemotherapy, radiation therapy, concurrent chemoradiation therapy, sorafenib, operation, or conservative care) were administered according to the clinician’s judgements.

### 4.4. Conventional Risk Prediction Models for HCC 

The ART, ABCR, SNACOR, HAP, and mHAP II models were devised to predict OS in patients with HCC undergoing TACE. The ART score features an increase in the AST level of >25%, an increase in the Child–Pugh score of 1 or 2 points from baseline, and the absence of a radiological tumor response [16]. The ABCR score features baseline AFP level (>200 ng/mL) and BCLC stage, a rise in the Child–Pugh score of ≥2 points from baseline, and the absence of a radiological Response [18]. The SNACOR model features tumor size (≥5 cm), tumor number (≥4), the baseline AFP level (≥400 ng/mL), Child–Pugh class B status, and the absence of an objective radiological response [17]. The HAP score features albumin (<36 g/dL), AFP level (>400 ng/mL), bilirubin (>17 μmol/L), and maximal tumor size (>7 cm) [19]. The mHAP II scores features albumin (<36 g/dL), AFP level (>400 ng/mL), bilirubin (>17 μmol/L), maximal tumor size (>7 cm), and tumor number (≥2) [20]. Higher scores are correlated independently with poor OS.

### 4.5. Statistical Analysis 

Data are expressed as medians (ranges) or frequencies (percentages), as appropriate. Student’s *t*-test (or the Mann–Whitney test, when appropriate) was used to compare continuous variables, and the chi-squared test (or Fisher’s exact test, when appropriate) was used to compare categorical variables. The Kaplan–Meier method was used to investigate survival; differences between groups were examined with the aid of the log-rank test.

Variables including components of the conventional risk prediction models were evaluated using univariate and multivariate Cox regression analyses. Variables with *p* < 0.05 in the univariate analysis were included as candidate variables for the multivariate Cox proportional-hazards models to identify a set of predictors to construct the predictive model; the β regression coefficients, *p*-values, and adjusted hazard ratios (HRs) with 95% confidence intervals (CIs) of all predictors were calculated. We calculated time-dependent areas under receiver-operating characteristic curves (AUROCs) for the ART, ABCR, SNACOR, HAP, and mHAP II models, and for our new SNAVCORN score (tumor size, tumor number, AFP level, vascular invasion, Child–Pugh score, objective response after TACE, neutrophil-to-lymphocyte ratio) model, to predict OS at 1, 3, and 5 years in the derivation set; the AUROCs were compared between model pairs using the method of Delong et al. The time-dependent AUROCs of OS risk prediction models at 1, 3, and 5 years for the derivation, internal validation, and external validation sets were calculated to estimate predictive performance. Statistical analyses were performed using IBM SPSS (ver. 20.0; IBM Co., Armonk, NY, USA) and R (http://cran.r-project.org/) software; two-sided *p*-values <0.05 were considered to reflect significance.

## 5. Conclusions

In conclusion, a high NLR is a simple, objective, and easily calculable biochemical marker predicting OS in patients with HCC undergoing TACE. Our SNAVCORN model based on the NLR and other predictors showed remarkable predictive performance. The SNAVCORN score stratifies patients with HCC undergoing TACE by prognosis and will aid in determining future treatment.

## Figures and Tables

**Figure 1 cancers-11-00509-f001:**
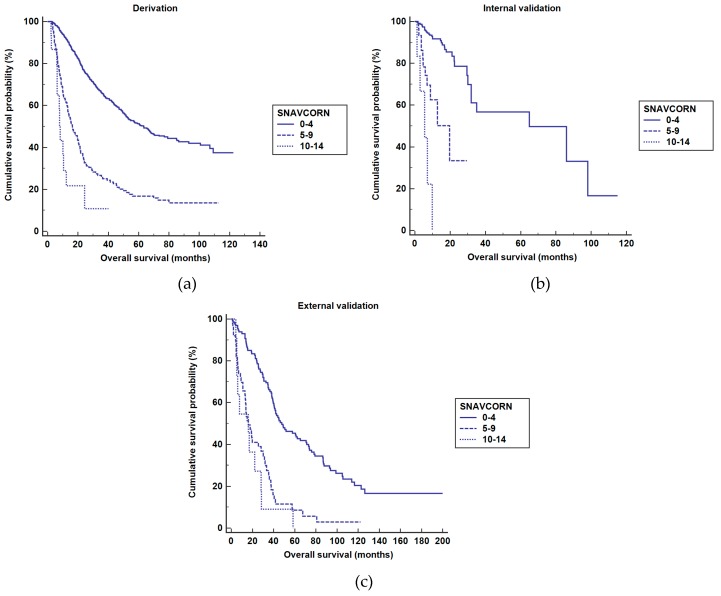
Stratification of patients according to SNAVCORN score (**a**) Derivation set. The median OS was significantly higher in the low-risk group at 63.3 (95% CI 52.7–74.5) months, followed by the intermediate-risk (16.3 (95% CI 12.0–20.6) months) and high-risk (8.3 (95% CI 7.0–9.6) months) groups (log-rank test, *p* < 0.001). (**b**) Internal validation set. The median OS was significantly higher in the low-risk group at 65.0 (95% CI 21.6–108.4) months, followed by the intermediate-risk (19.6 (95% CI 7.6–31.5) months) and high-risk (5.6 (95% CI 0.5–10.7) months) groups (log-rank test, *p* < 0.001). (**c**) External validation set. The median OS was significantly higher in the low-risk group at 47.7 (95% CI 32.5–62.9) months, followed by the intermediate-risk (16.2 (95% CI 9.9–22.4) months) and high-risk (7.1 (95% CI 3.5-10.7) months) groups (log-rank test, *p* < 0.001).

**Table 1 cancers-11-00509-t001:** Baseline characteristics of study population.

Variables	Derivation	Internal Validation	External Validation	*p*-Value ^a^	*p*-Value ^b^
	(n = 921)	(n = 395)	(n = 381)		
Age (years)	68.2 ± 10.7	65.9 ± 10.4	59.6 ± 11.3	<0.001	<0.001
Male gender	700 (76.0)	317 (80.3)	292 (76.6)	0.099	0.83
Etiologies					
HBV	648 (70.4)	267 (67.6)	269 (70.6)	0.617	0.047
HCV	128 (13.9)	55 (13.9)	42 (11.0)		
Alcohol	133 (14.4)	70 (17.7)	50 (13.1)		
Others	12 (1.3)	3 (0.8)	20 (5.2)		
AFP (ng/mL)	51 (6–283)	67 (5–480)	46 (7–123)	0.193	0.149
Tumor size (cm)	5.0 (2.0–8.2)	5.1 (3.0–8.4)	3.5 (1.6–6.2)	0.413	0.042
Tumor number					
Single	683 (74.2)	320 (81.0)	256 (67.2)	0.066	<0.001
2 or 3	101 (10.9)	30 (7.6)	59 (15.5)		
≥4	137 (14.9)	45 (11.4)	66 (17.3)		
Vascular Invasion	204 (22.1)	90 (22.8)	64 (16.8)	0.488	0.072
BCLC stage					
A	425(46.2)	199 (50.4)	237 (62.2)	0.418	<0.001
B	352(38.2)	153 (38.7)	93 (24.4)		
C	144(15.6)	43 (10.9)	51 (13.4)		
Performance status				0.324	0.146
0	807 (88.6)	361 (91.4)	353 (92.7)		
1–2	114 (12.4)	34 (8.6)	28 (7.3)		
Child–Pugh class					
A	813 (88.3)	354 (89.6)	303 (79.5)	0.508	<0.001
B	108 (11.7)	41 (10.4)	78 (20.5)		
Treatment response					
Complete response	465 (50.4)	170 (43.0)	152 (39.9)	0.080	0.058
Partial response	262 (28.5)	150 (38.1)	125 (32.8)		
Stable disease	115 (12.5)	35 (8.9)	16 (4.2)		
Progressive disease	79 (8.6)	40 (10.0)	88 (23.1)		
AST (IU/L)	57 ± 57	45 ± 33	63 ± 65	<0.001	0.129
ALT (IU/L)	41 ± 43	34 ± 29	45 ± 43	0.003	0.418
Serum albumin (g/dL)	3.7 ± 0.5	3.8 ± 0.5	3.7 ± 0.6	0.185	0.108
Total bilirubin (g/dL)	0.9 ± 0.8	0.9 ± 0.5	1.0 ± 1.0	0.365	0.117
Prothrombin time (INR)	1.1 ± 0.4	1.2 ± 0.4	1.2 ± 0.2	0.516	0.128
Platelet (10^9^/L)	145 ± 75	140 ± 69	129 ± 72	0.409	0.546
NLR ratio	3.5 ± 4.3	2.8 ± 2.4	2.3 ± 1.7	<0.001	<0.001

Variables are expressed as median (range), mean ± (SD), or n (%). *p*-value ^a^: Derivation vs. Internal validation; *p*-value ^b^: Derivation vs. External validation. HBV, hepatitis B virus; HCV, hepatitis C virus; AFP, alpha-feto protein; BCLC, the Barcelona clinic liver cancer; AST, aspartate minotransferase; ALT, alanine transaminase; NLR, Neutrophil-to-lymphocyte ratio.

**Table 2 cancers-11-00509-t002:** Prognostic factors for overall survival and risk score calculation in the derivation cohort (n = 921).

Variables	Univariate	Multivariate			
		*p*-Value	Adjusted HR (95% CI)	β-Coefficient	SNAVCORN Risk Score
Tumor size	<0.001	<0.001			
≤3cm			1.000		0
>3cm			1.555 (1.252–1.932)	0.441	1
Tumor number	<0.001	<0.001			
≤3			1.000		0
>3			1.752 (1.354–2.268)	0.561	2
AFP	<0.001	0.032			
<200			1.000		0
≥200			1.291 (1.022–1.629)	0.255	1
Vessel invasion	<0.001	<0.001			
No			1.000		0
Present			2.272 (1.793–2.879)	0.821	3
Child–Pugh score	<0.001	<0.001			
5			1.000		0
6			1.483 (1.195–1.840)	0.394	2
7–8			1.903 (1.412–2.563)	0.643	3
9			2.852 (1.401–5.806)	1.048	4
Objective response	<0.001	<0.001			
CR+PR			1.000		0
SD+PD			1.663 (1.267–2.182)	0.508	2
NLR ratio	<0.001	0.015			
<5			1.000		0
≥5	<0.001	0.015	1.380 (1.064–1.789)	0.322	1

AFP, alpha-feto protein; CR, complete response; PR, partial response; SD, stable disease; PD, progressive disease; NLR, neutrophil-to-lymphocyte ratio.

**Table 3 cancers-11-00509-t003:** Predictive performances of SNAVCORN score to predict overall survival.

Year	Time-Dependent AUROC (95% CI)		
	Derivation Set (n = 921)	Internal Validation Set (n = 395)	External Validation Set (n = 381)
Year 1	0.812 (0.769–0.856)	0.868 (0.795–0.942)	0.801 (0.716–0.885)
Year 3	0.734 (0.396–0.770)	0.742 (0.650–0.833)	0.789 (0.726–0.853)
Year 5	0.700 (0.663–0.737)	0.745 (0.658–0.832)	0.725 (0.655–0.796)

AUROC, area under receiver-operating characteristic curve; CI, confidence interval. SNAVCORN score features baseline tumor size (≥5 cm), tumor number (≥4), AFP level (≥400 ng/mL), presence of vascular invasion, Child–Pugh score (≥6), the absence of objective response after TACE, and NLR (≥5).

**Table 4 cancers-11-00509-t004:** Predictive performances of prediction models in derivation set (n = 921).

Time-Dependent AUROC (95% CI)						
**Year**	**SNAVCORN**	**ART**	**SNACOR**	***p*-value**	***p*-value**	
				**SNAVCORN vs. ART**	**SNAVCORN** **vs. SNACOR**	
Year 1	0.812 (0.769–0.856)	0.588 (0.531–0.645)	0.762 (0.707–0.818)	<0.001	<0.001	
Year 3	0.734 (0.396–0.770)	0.503 (0.462–0.545)	0.662 (0.621–0.704)	<0.001	<0.001	
Year 5	0.700 (0.663–0.737)	0.490 (0.450–0.529)	0.634 (0.593–0.675)	<0.001	<0.001	
	**ABCR**	**HAP**	**mHAPII**	***p*-value**	***p*-value**	***p*-value**
				**SNAVCORN vs. ABCR**	**SNAVCORN** **vs. HAP**	**SNAVCORN** **vs. mHAPII**
Year 1	0.786 (0.728–0.844)	0.744 (0.698–0.790)	0.705 (0.655–0.756)	<0.001	<0.001	<0.001
Year 3	0.667 (0.609–0.725)	0.688 (0.650–0.725)	0.658 (0.619–0.696)	0.003	<0.001	<0.001
Year 5	0.617 (0.556–0.677)	0.644 (0.627–0.702)	0.650 (0.612–0.687)	<0.001	<0.001	<0.001

AUROC, area under receiver-operating characteristic curve; CI, confidence interval. SNAVCORN score features baseline tumor Size (≥5 cm), tumor Number (≥4), AFP level (≥400 ng/mL), presence of Vascular invasion, Child–Pugh score (≥6), the absence of Objective Response after TACE, and NLR (≥5); ART score features an increase in the AST level of >25%, an increase in the Child–Pugh score of 1 or 2 points from baseline, and the absence of a radiological tumor response; ABCR score features baseline AFP level (>200 ng/mL) and BCLC stage, a rise in the Child–Pugh score of ≥2 points from baseline, and the absence of a radiological Response; SNACOR model features tumor Size (≥5 cm), tumor Number (≥4), the baseline AFP level (≥400 ng/mL), Child–Pugh class B status, and the absence of an Objective radiological Response; HAP score features albumin (<36 g/dL), AFP level (>400 ng/mL), bilirubin (>17 μmol/L), and maximal tumor size (>7 cm); mHAP II score features albumin (<36 g/dL), AFP level (>400 ng/mL), bilirubin (>17 μmol/L), maximal tumor size (>7 cm), and tumor number (≥2).

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
