# Peer review of "Development of a New Nomogram Including Neutrophil-to-Lymphocyte Ratio to Predict Survival in Patients with Hepatocellular Carcinoma Undergoing Transarterial Chemoembolization"

_cancers, 2019, doi:10.3390/cancers11040509_

Round 1
Reviewer 1 Report
In this manuscript, Chon et al. reported that ABCRN score, which is calculated from AFP level, BCLC stage, Child–Pugh class, treatment Response after TACE, and NLR, provides better prediction of the prognosis of patients with HCC after TACE than other conventional predictors. The study was well designed, and the number of patients were sufficient to demonstrate its accuracy as a predictor. The results were clear and statistically robust. Thus, this paper demonstrated the clinical usefulness of ABCRN.
The authors should address the following comments, which may be helpful for improving the manuscript.
1. Although the authors compared AUROC with other scores, the comparison of the Kaplan-Meier analysis also provides additional insightful information for readers.
2. As the authors described in the Discussion, it is also important to accurately predict the patients who will benefit from TACE, before the treatment. The authors should assess and discuss the possibility of prediction without "R" (prediction with ABCN).
Author Response
Comment#1
Although the authors compared AUROC with other scores, the comparison of the Kaplan-Meier analysis also provides additional insightful information for readers
Response) Thank you for the good comment. We added the Kaplan-Meier curves of other scores in the supplementary Fig 1.
Comment#2
As the authors described in the Discussion, it is also important to accurately predict the patients who will benefit from TACE, before the treatment. The authors should assess and discuss the possibility of prediction without "R" (prediction with ABCN).
Response) Thank you for the reviewer’s opinion. As Reviewer 2 suggested us to build up a new model including various baseline variables, we invested SNAVCORN model. Therefore, we analyzed whether including TACE response to baseline factors further enhanced the predictive performance in SNAVCORN model. SNAVCN score consisting of only six baseline predictors was simpler, however, addition of the TACE response (SNAVCORN) significantly improved predictive performance. We showed this result in supplementary Table 2 and mentioned it in discussion section.

Reviewer 2 Report
This is an interesting multicenter retrospective study proposing a new prognostic score to predict the survival outcome of HCC patients undergoing TACE.
The study design and the results of this study seem original (introduction of NLR in a prognostic score for TACE) and interesting. However, I have found some major issues in reviewing this paper:
1) The Materials and Methods section seems poor. It is not clear how many TACE procedures have been performed for each patient, how sequential therapies have been recorded (i.e. Sorafenib after TACE), the follow up / treaement strategy used according to response to first TACE. The statistical analysis paragraph is too short. How variables for multivariable analsysis were selected?
2) In the Results section again it is not clear which therapies patients performed after first TACE and how many TACE procedures for patient have been performed.
3) The authors included Child Classes and BCLC stages in multivariable analysis. However Child A and B classes are prognostically extremely heterogeneous. For example Child Pugh A5 is different from 6, and Child Pugh B7 is different from 8 and 9.
I suggest to the authors to use Child Pugh score and not Child Classes as prognostic variable.
Similarly BCLC stages definition has several bias particularly in a specific population undergoing a single main therapy. Probably the prognostic performance of the final score would be better including simple BCLC variables in the score (i.e. nodule diameter and number, vascular invasion, performance status).
4) I suggest to perform a univariate survival analysis exploring all simple variables (i.e. nodule diameter and number, vascular invasion, performance status) and to select variables with p<0.1 for the multivariable model. I suspect that the AUROC values of this kind of new model would be better than that of the ABCRN score.
5) I suggest to compare the final model also to the HAP score, mHAPII, and mHAPIII scores.
Author Response
Comment #1
The Materials and Methods section seems poor. It is not clear how many TACE procedures have been performed for each patient, how sequential therapies have been recorded (i.e. Sorafenib after TACE), the follow up / treatment strategy used according to response to first TACE. The statistical analysis paragraph is too short. How variables for multivariable analsysis were selected?
Response) We agree with the reviewer’s opinion, and tried to dictate in detail about the sequential procedures after first TACE in the Method section. In patients who showed complete response at CT scan 1 month after first TACE, were regularly followed with CT scan taken every 3-6 month. In patients showing PR, SD, or PD at response evaluation CT scan 1 month after first TACE, subsequent treatments (i.e. repeated TACE, radio frequency ablation, intraarterial chemotherapy, radiation therapy, concurrent chemoradiation therapy, sorafenib, operation, or conservative care) were administered according to the clinician’s judgements.
Variables including components of the conventional risk prediction models were evaluated using univariate and multivariate Cox regression analyses. Variables with P<0.05 in the univariate analysis were included as candidate variables for the multivariate Cox proportional-hazards models to identify a set of predictors to construct the predictive model; the β regression coefficients, p-values, and adjusted hazard ratios (HRs) with 95% confidence intervals (CIs) of all predictors were calculated. We added this in the Method section.
Comment #2
In the Results section again it is not clear which therapies patients performed after first TACE and how many TACE procedures for patient have been performed.
Response) Thank you for the reviewer’s opinion and we added in the result section as follows: In 262 patients who showed PR after first TACE, 121 (46.2%) patients underwent repeated TACE, 95 (36.2%) patients underwent concurrent chemoradiation therapy, 12 (4.6%) patients underwent radiation therapy, 8 (3.1%) patients underwent sorafenib, 8 (3.1%) patients underwent liver resection, 8 (3.1%) patients underwent radio frequency ablation (RFA), 8 (3.1%) patients underwent intra-arterial chemotherapy, and 2 (0.8%) patients had conservative care. Among all patients with PR, 76 (29%) patients achieved CR after this additional therapy. In 194 patients with who showed SD or PD after 1st TACE, 66 (34.0%) patients underwent repeated TACE, 60 (30.9%) patients underwent concurrent chemoradiation therapy, 21 (10.8%) patients underwent sorafenib, 20 (10.3%) patients had conservative care, 12 (6.2%) patients underwent radiation therapy, 11 (5.7%) patients underwent intra-arterial chemotherapy, and 4 (2.1%) patients had RFA. Patients had undertaken mean of 1.4 sessions of TACE for the tumors. We added this in the result section.
Comment #3
The authors included Child Classes and BCLC stages in multivariable analysis. However Child A and B classes are prognostically extremely heterogeneous. For example Child Pugh A5 is different from 6, and Child Pugh B7 is different from 8 and 9. I suggest to the authors to use Child Pugh score and not Child Classes as prognostic variable. Similarly BCLC stages definition has several bias particularly in a specific population undergoing a single main therapy. Probably the prognostic performance of the final score would be better including simple BCLC variables in the score (i.e. nodule diameter and number, vascular invasion, performance status).
Response) Thank you for the valuable comment. We found out that patients with Child-Pugh score 5, those with Child-Pugh score 6, those with Child-Pugh score 7-8, and those with Child-Pugh score 9 showed different prognoses. According to the review’s opinion, we incorporated single factors of BCLC stages (tumor diameter, tumor number, and vascular invasion) and child-Pugh score (instead of Child-Pugh Class) in to the new model. A new SNAVCORN score (tumor Size, tumor Number, AFP level, Vascular invasion, Child-Pugh score, Objective Response after TACE, Neutrophil-to-lymphocyte ratio) showed higher prognostic performance in terms of higher AUROC than ABCRN score and other conventional prediction models. We changed this in the Result section accordingly.
* Predictive performances of ABCRN vs. SNAVCORN score to predict overall survival | |||
Time-dependent AUROC (95% CI) | P-value | ||
Year | ABCRN | SNAVCORN | |
Year1 | 0.800 (0.748-0.852) | 0.812 (0.769-0.856) | <0.001 |
Year3 | 0.720 (0.673-0.766) | 0.734 (0.396-0.770) | <0.001 |
Year5 | 0.683 (0.635-0.732) | 0.700 (0.663-0.737) | <0.001 |
AUROC, area under receiver-operating characteristic curve; CI, confidence interval. | |||
Comment #4
I suggest to perform a univariate survival analysis exploring all simple variables (i.e. nodule diameter and number, vascular invasion, performance status) and to select variables with p<0.1 for the multivariable model. I suspect that the AUROC values of this kind of new model would be better than that of the ABCRN score.
Response) According to the reviewer’s opinion, we performed univariate survival analyses using various baseline factors (Supple Table 1), and selected variables to invest a new model. The new SNAVCORN score (tumor Size, tumor Number, AFP level, Vascular invasion, Child-Pugh score, Objective Response after TACE, Neutrophil-to-lymphocyte ratio) showed higher prognostic performance in terms of higher AUROC than ABCRN score.
Comment #5
I suggest to compare the final model also to the HAP score, mHAPII, and mHAPIII scores.
Response) We compared the prognostic performance of new SNAVCORN score with the HAP score and mHAPII scores. Because the numerical values of mHAP III were automatically calculated according to the computer-defined formula, we could not compare the prognostic performance. We added this in the Result section and in Table 4.

Round 2
Reviewer 1 Report
The authors sincerely and adequately addressed my comments.
Now, this manuscript is convincingly recommended for publication in the Cancers.
Reviewer 2 Report
None